# The Effects of Sheep Tail Fat, Fat Level, and Cooking Time on the Formation of Nε-(carboxymethyl)lysine and Volatile Compounds in Beef Meatballs

**DOI:** 10.3390/foods12152834

**Published:** 2023-07-26

**Authors:** Kübra Öztürk, Zeynep Feyza Yılmaz Oral, Mükerrem Kaya, Güzin Kaban

**Affiliations:** 1Department of Food Engineering, Faculty of Agriculture, Atatürk University, Erzurum 25240, Türkiye; ozturkkubra1516@gmail.com (K.Ö.); mkaya@atauni.edu.tr (M.K.); 2Department of Food Technology, Erzurum Vocational School, Atatürk University, Erzurum 25240, Türkiye; zeynep.yilmaz@atauni.edu.tr; 3MK Consulting, Ata Teknokent, Erzurum 25240, Türkiye

**Keywords:** meatball, CML, AGE, sheep tail fat, beef fat, volatile compounds

## Abstract

This study aimed to determine the effects of fat type (sheep tail fat (STF) and beef fat (BF)), fat levels (10, 20, or 30%), and cooking time (0, 2, 4, and 6 min, dry heat cooking at 180 °C) on the carboxymethyl lysine (CML) content in meatballs. pH, thiobarbituric acid reactive substance (TBARS), and volatile compound analyses were also performed on the samples. The use of STF and the fat level had no significant effect on the pH value. The highest TBARS value was observed with the combination of a 30% fat level and STF. CML was not affected by the fat level. The highest CML content was determined in meatballs with STF at a cooking time of 6 min. In the samples cooked for 2 min, no significant difference was observed between STF and BF in terms of the CML content. STF generally increased the abundance of aldehydes. Aldehydes were also affected by the fat level and cooking time. A PCA provided a good distinction between groups containing STF and BF regardless of the fat level or cooking time. Pentanal, octanal, 2,4-decadienal, hexanal, and heptanal were positively correlated with CML.

## 1. Introduction

The Maillard reaction (MR) is initiated by the condensation of amino groups on proteins, peptides, and amino acids with carbonyl groups on reducing sugars. The main pathways of the MR involve three steps: initial, intermediate or advanced, and final stages [1]. The initial stage of MR consists of a condensation of the reducing saccharide with the amino compound, resulting in the formation of an Amadori product and a Heyns product. In the intermediate/advanced stage of the MR, Amadori and Heyns products can undergo degradation reactions, and several α-dicarbonyl compounds are formed. These compounds are also involved in further reactions with side chains of peptides or proteins, leading to the formation of advanced glycation end products (AGEs). In the final stage of the MR, nitrogen-containing brown polymers or copolymers called melanoidins are formed by the condensation and polymerization reactions of previously formed reactive intermediates [1,2,3]. The MR is influenced by many factors such as the physical state of the matrix, pH, the type and concentration of reactants, process conditions, the temperature, duration of storage, and water activity [1,4].

The MR contributes significantly to quality characteristics of food products, including flavor, aroma, and color. In addition, mutagenic and toxigenic compounds can also be formed as a result of this reaction. Furthermore, the MR can lead to the loss of the nutritional value of proteins [2]. AGEs formed during the Maillard reaction are heterogeneous compounds, and these compounds have adverse effects on human health and are known to be closely related to many chronic diseases [5,6,7,8]. For example, it has been reported that an AGE-rich diet can lead to increased oxidative stress and inflammation linked with type 2 diabetes [9].

AGEs are compounds that occur naturally in foods of animal origin. The AGE level can vary depending on many factors such as the source and composition of meat, the heat treatment process and duration, and the protein and fat contents [5,10,11,12]. Although AGEs such as N-ε-carboxymethyl-lysine (CML), methyl glyoxal lysine dimer, N-ε-carboxyethyl-lysine (CEL), and pentosidine are commonly found in foods and particularly in processed meat products, CML is established as an indicator of AGEs due to its resistance to acidic food environments [7,11]. The cooking technique applied to meat and meat products, the cooking degree, and the fat content play an important role in CML formation [7].

Minced meat products, such as meatballs, burgers, and meat patties, are widely consumed around the world. They are produced using different methods depending on the type of meat, cost considerations, the shape, the nutritional value, and religious reasons [13]. In these products, fat is a major ingredient in terms of taste, texture, and flavor. In addition to beef fat, sheep tail fat is also used in meatball production. Sheep tail fat contains more unsaturated fatty acids than beef fat (intermuscular fat) [14,15,16]. Sheep tail fat also contains relatively higher levels of polyunsaturated fatty acids (PUFA) [17,18,19] and nutraceutical fatty acids (n-3 polyunsaturated fatty acids), which have health-promoting benefits [20], than beef fat. It was reported that the contents of oleic acid (C18:1) and linoleic acid (C18:2) in sheep tail fat varied between 41.51 and 49.7 and between 2.62 and 5.7%, respectively. However, it was stated that the fatty acid composition is affected by breed, age, sex, and nutritional conditions [21]. On the other hand, the fact that sheep tail fat is more susceptible to oxidation is of great importance in determining the effect of this fat on CML formation. In addition, it is important to determine the effect of sheep tail fat on the volatile profile and also to reveal the relationship between volatile compounds and CML in meatballs.

There are limited studies on CML formation in meatballs. In these studies, the effects of adding different amounts of salt to beef patties [22], the storage time of frozen pork patties [23], adding various proportions of wheat, rye, and triticale bran to beef patties [5], and the use of *Kaempferia galanga* L. and kaempferol extracts [24] on CML formation were investigated. However, there is no information on the effects of sheep tail fat and fat levels on CML formation in beef meatballs. The aim of this study is to determine the influences of fat type (beef fat and sheep tail fat), fat level (10, 20, and 30%), and cooking time (0, 2, 4, and 6 min at 180 °C) on CML formation in meatballs. In addition, the effects of these factors on pH, lipid oxidation, and volatile compounds were also investigated.

## 2. Materials and Methods

### 2.1. Chemicals, Reagents and Standards

Trichloroacetic acid (TCA), ethylenediaminetetraacetic acid (EDTA), propyl gallate, and sodium borate were purchased from Sigma Aldrich (Steinheim, Germany), and thiobarbituric acid (TBA) and sodium borohydride were purchased from Merck (Darmstadt, Germany). Chromatography-HPLC grade solvents, including chloroform, methanol, and acetonitrile, were purchased from Fisher Scientific (Schwerte, Germany), J.T. Baker (Gliwice, Poland), and Sigma Aldrich (Steinheim, Germany), respectively. The carboxymethyl-lysine standard was obtained from Cayman Chemical (Ann Arbor, MI, USA). The standard substances for volatile compound analysis, ethyl acetate, pentanal, hexanal, heptanal, octanal, nonanal, 2,4-decadienal, 2-heptanone, 1-pentanol, 1-hexanol, and 1-heptanol were purchased from Merck (Hohenbrunn, Germany), and the standard mix was purchased from Supelco (parrafine mix, 44585-U, Bellefonte, PA, USA).

### 2.2. Material

In the study, lean beef, beef fat (intermuscular fat), and sheep tail fat were used as raw materials. The meat was obtained from the round part of the cattle carcasses by a local butcher. Beef fat and sheep tail fat also came from the same butcher. Since the research was conducted in three replications, meat and fat were procured at three different times.

### 2.3. Meatball Production and Cooking

Lean beef, beef fat, and sheep tail fat were minced separately through a 3 mm plate using a meat mincer (MADO Typ MEW 717, Dornhan, Germany). The experiment was carried out according to a 2 × 3 × 4 factorial design with two types of fat (beef fat and sheep tail fat), three levels of fat (10%, 20%, and 30%), and four levels of cooking time (0, 2, 4, and 6 min at 180 °C on a hot plate). Six meatball patties were prepared based on fat type and fat level and mixed by hand and processed into meatballs (1.5 cm thick and 7.5 cm diameter) using a metal shaper. The weight of the meatballs was 50 g. Salt was added to the mix at a rate of 1.5%. Three independent experiments were carried out, and thus a total of 18 meatball mixes were prepared.

Each group was divided into four subgroups. The first group of meatballs were evaluated as the control group (raw–uncooked). The second, third, and fourth group of meatballs were cooked for 2 min (1 min per side), 4 min (2 min per side), and 6 min (3 min per side), respectively. The cooking process was carried out on a hot plate preheated at 180 °C and the temperature was controlled using a thermocouple.

### 2.4. Pysicochemical Analysis

After the cooking process, the samples were cooled down to room temperature and then homogenized with a blender to obtain a uniform sample for the analyses. The homogenized samples were stored at −18 °C until analysis.

#### 2.4.1. pH and TBARSs

For pH measurements, 10 g of homogenized sample was weighed and 100 mL of distilled water was added to it. After homogenizing with an ultra-turrax (IKA Werk T 25, Staufen, Germany) for 1 min, the pH value was determined by a pH meter (Mettler Toledo, Greifensee, Switzerland). The pH meter was calibrated with buffer solutions (pH 4.0 and pH 7.0) before use, and thermal compensation was carried out automatically [25].

In TBARS analyses, 2 g of homogenized sample was mixed with a 12 mL TCA solution (7.5% TCA, 0.1% EDTA, and 0.1% propyl gallate). After homogenization, it was filtered through a Whatman 1 filter paper and 3 mL of filtrate was added to a 0.02 M TBA solution (0.02 M). Afterwards, samples were kept in boiling water bath for 40 min, then centrifugation (Beckman Coulter, Allegra X-30R, Indianapolis, IN, USA) was applied for 5 min at 2000 G. The absorbance of the samples was determined at 530 nm, and the results were expressed as μmol MDA/kg [26].

#### 2.4.2. Nε-(carboxymethyl)lysine (CML)

CML analysis was performed according to the method given by Chen and Smith [11]. After a 0.2 g sample was added to 20 mL of chloroform/methanol (2:1, *v*/*v*), it was centrifuged (10,000× *g* at 4 °C) (Beckman Coulter, Brea, CA, USA) to remove the fat. After incubation for 4 h with 4 mL sodium borohydride (1 M in 0.1 N NaOH) and 8 mL sodium borate buffer (0.2 M, pH 9.4), samples were hydrolyzed with 6 mL 12 M HCl at 110 °C for 20 h. Then, samples were dried using rotary evaporation and were added to 10 mL of water and dissolved in 10 mL of sodium borate buffer, followed by a final filtration. Prior to HPLC analysis, a 50 μL extract was mixed with 200 μL of ortho-phtalaldehyde derivatization reagent for 5 min. The CML amount was determined using HPLC (Agilent 1100, Santa Clara, CA, USA) with a fluorescence detector (Agilent, Santa Clara, CA, USA). The determination was performed with a reverse phase TSK gel ODS-80 TM column (25 cm × 4.6 mm, 5 μm, Tosohass, Montgomeryville, PA, USA) and with the fluorescence settings of 340 nm (excitation) and 455 nm (emission). The flow rate was 1.0 mL/min and the injection volume was 20 μL. Acetate buffer/acetonitrile (90:10, *v*/*v*) and acetonitrile were used as mobile phases, and the flow of the mobile phase was gradually changed. The recovery rates were determined by spiking cooked meat with N-ε-(carboxymethyl)-lysine standard at six different levels (1–30 μg/mL). Each treatment was replicated five times. The mean recoveries ranged from 102.20% to 104.46%, with relative standard deviations between 1.10% and 2.18%. The regression line coefficient (R^2^) for CML was 0.999. The limit of detection (LOD) and the limit of quantification (LOQ) were calculated using dilutions of the standard solution according to the following formulas: LOD = 3.3 × Sy/s and LOQ = 10 × Sy/s. The LOD and LOQ for CML were 0.64 and 1.95 μg/mL, respectively.

#### 2.4.3. Volatile Compounds

An amount of 5 g of homogenized sample was placed into a 40 mL vial (Supelco, Bellefonte, PA, USA). The extraction of volatile compounds was performed using solid phase microextraction (SPME) with carboxen/polydimethylsiloxane fiber (CAR/PDMS, 75 µm, Supelco, Bellefonte, PA, USA). The sample was kept at 30 °C for 1 h in a thermal block (Supelco, Bellefonte, PA, USA) to collect the volatile compounds. After equilibration, the SPME fiber was exposed to the sample headspace at 30 °C for 2 h. Gas chromatography/mass spectrometry (Agilent, Santa Clara, CA, USA) was used to identify volatile compounds. A DB-624 (J&W Scientific, 30 m × 0.25 mm × 1.4 μm film) was used as the column and the carrier gas was helium. The oven temperature was first set to 40 °C for 6 min, then gradually increased to 210 °C and then held at 210 °C for 12 min. The injector port was in splitless mode. The GC/MS interface was maintained at 280 °C. Mass spectra were obtained by electron impact at 70 eV, and the quadrupole mass spectrometer scan range was 40–400 atomic mass units.

Mass spectrometry libraries (NIST, FLAVOR, and WILEY) and standard substances were used for the identification of compounds, and the Kovats index was determined using the standard mix (Supelco 44585-U). The results are given as AU × 10^6^ [27].

### 2.5. Statistical Analysis

Data were analyzed by an analysis of variance (ANOVA) using a general linear model considering the fat type (beef fat and sheep tail fat), fat level (10, 20, and 30%), and cooking time (0, 2, 4, and 6 min) as the main effects, and the replicates as a random effect for a randomized complete block design. The experiment was repeated 3 times and each experiment was carried out at different times using different raw materials. The differences between the means were determined using Duncan’s multiple range tests at the *p* < 0.05 level. All statistical analyses were performed using SPSS version 20 statistical program (SPSS Inc., Chicago, IL, USA). In addition, principal component analysis (PCA) was performed to determine the relationship between fat type, fat level, and cooking time for volatile compounds as well as CML content using Unscrambler software (CAMO version 10.1, Oslo, Norway). The differential profile (cluster heat map) between the factors, volatile compound groups, and CML was also analyzed using heat mapper (http://www.heatmapper.ca, accessed on 1 June 2023).

## 3. Results and Discussion

### 3.1. pH and TBARSs

The effects of fat type, fat level, and cooking time on pH, TBARS value, and the CML content of meatballs are given in Table 1. The fat type and fat level had no significant effect on the pH value of samples (*p* > 0.05) (Table 1). In contrast, the cooking time had a very significant impact (*p* < 0.01) on the pH, and the mean pH value increased with increasing cooking times. However, no significant differences in pH value were observed between 4 and 6 min of cooking. Furthermore, the interactions of all factors were insignificant (*p* > 0.05) (Table 1). The increase in pH value with cooking can be explained by the reduction of carboxylic groups on proteins and also by the release of calcium and magnesium ions from proteins [28].

Processed meat products are more sensitive to lipid oxidation than fresh meat due to mincing and heat treatment. Malondialdehyde is evaluated to be the most significant degradation product, arising from lipid oxidation, and a TBARS analysis is widely used in its determination [29]. The TBARS value was affected by the fat type (*p* < 0.01), and the highest mean TBARS value was observed in the group with sheep tail fat (Table 1). It is thought that this result is due to the fact that sheep tail fat contains more unsaturated fatty acids than beef fat and therefore is more sensitive to oxidation [14]. In addition, the fat level (*p* < 0.05) and cooking time (*p* < 0.01) had an effect on TBARSs. The highest TBARS value was found for the groups containing 30% fat. In addition, the mean TBARS value increased with increasing cooking time (Table 1). There is no legal limit for the MDA level determined in the TBARS analysis applied to determine the degree of lipid oxidation in meat products [30]. However, it is suggested that when the TBARS value reaches 1 mg MDA/kg, it can create malodor and it can be detected [31]. In this study, in all groups except for 6 min of cooking time, the TBARS value was below 1 mg MDA/kg.

As shown in Figure 1, the fat type × fat level interaction was determined. The TBARS value for beef fat was similar at all fat levels. On the other hand, meatballs prepared with sheep tail fat had the highest TBARS value at 30% fat level, and the lowest TBARS value at 10% fat level. Furthermore, differences between fat types were not significant at 10% and 20% fat levels, and at the 30% fat level, sheep tail fat had a higher mean TBARS value than beef fat. According to these results, 30% sheep tail fat significantly increased lipid oxidation (Figure 1). This result is likely due to the fact that sheep tail fat contains more unsaturated fatty acids than beef fat [14].

### 3.2. Nε-(carboxymethyl)lysine (CML)

Fat type had a very significant (*p* < 0.01) effect on CML content. Cooking time also showed an effect at the *p* < 0.01 level on CML content (Table 1). Sheep tail fat has a higher CML content than beef fat. At the same time, sheep tail fat showed a higher TBARS value, an indicator of lipid oxidation, than beef fat (Table 1). These results show a positive relationship between CML formation and lipid oxidation. It has also been determined in some other studies conducted on meat products that lipid oxidation increases CML formation [7,32]. In addition, in a study performed on fresh cooked meats, it was reported that irradiation accelerated lipid oxidation and thus enhanced CML formation [33]. Yu et al. [34], in their study on raw and heat-treated meats, reported that there was a positive correlation between the TBARS value and CML and that high temperatures promoted AGE formation. In the present study, sheep tail fat, which is more susceptible to autoxidation than beef fat, caused a significant increase in the TBARS value and therefore accelerated the formation of CML.

In the present study, the CML content increased with the duration of cooking time. However, no statistically significant difference was found between the samples cooked for 2 min and 4 min. The highest mean CML content was observed after 6 min (Table 1). The cooking method, temperature, and time are important factors in the formation of CML in meat products. In addition, in a study conducted on ground beef, it was revealed that the CML content increased with the heat treatment time and temperature (65–100 °C and 0 to 60 min) [35]. Additionally, it was stated that in cooked meat products, factors such as the source and composition of the meat, protein, and fat content are effective in the formation of AGEs [10,11,12]. In the present study, the CML content of the meatballs had no effect depending on the fat level. However, Bayrak Kul et al. [36] reported that the fat level is an important factor in the formation of CML, and the lowest CML content was found when using 10% fat. It is believed that this difference is probably due to the product type and cooking conditions.

The CML content of meatballs was significantly (*p* < 0.05) affected by the interaction of fat type and cooking time (Figure 2). The CML level increased in meatballs prepared using beef fat after 2 min of cooking, and a prolonged cooking time did not affect the CML levels (Figure 2). On the other hand, the CML content increased with increasing cooking time in meatballs with sheep tail fat. However, there was no statistical difference in CML contents between 2 and 4 min of cooking time. In addition, it was observed that the fat type was not important in the raw and cooked samples after 2 min. In the case of prolonged cooking times, the groups containing sheep tail fat had a higher CML content (Figure 2). This result shows that sheep tail fat is more effective in CML formation when the heat treatment time is increased.

Although the mechanism of the Maillard reaction has not yet been fully explained [37,38], it is known that the reaction rate may vary depending on the amount of reducing sugar and free amino groups in the environment and the temperature [37]. In our study, it was also revealed that the CML level increased as the time increased in dry heat cooking at 180 °C.

### 3.3. Volatile Compounds

A total of 25 compounds including aliphatic hydrocarbons, esters, aldehydes, ketones, alcohols, and furans were identified (Table 2). The fat type was found to have a significant impact on ethyl acetate, heptanal, 2-nonenal, 2,4-decadienal, 2-butanone, 2-octanone, 2-nonanone, 1-pentanol, 1-hexanol, 1-octene-3-ol, and furan. In addition, butyl propionate, pentanal, hexanal, and 4-decenal were affected by the fat type at a level of *p* < 0.05. The fat level caused a statistical difference at the *p* < 0.05 or *p* < 0.01 levels for five compounds (pentanal, hexanal, heptanal, octanal, and 2-butanone). The cooking time, another factor, had a very significant or significant effects on ethyl acetate, pentanal, hexanal, heptanal, octanal, 2,4-decadienal, and 2-octanone (*p* < 0.05 or *p* < 0.01) (Table 2).

Ethyl acetate, butyl propionate, 2-butanone, 1-octen-3-ol, 1-hexanol, and 1-pentanol gave higher mean values in the groups containing beef fat compared to the groups containing sheep tail fat. On the other hand, the highest abundances were observed in meatballs with sheep tail fat for pentanal, hexanal, heptanal, 2-nonenal, 4-decenal, 2,4-decadienal, 2-octanone, 2-nonanone, and furan (Table 2). As can be seen from Table 2, aldehydes were higher in the group containing sheep tail fat than the group with beef fat. This result is due to the high content of polyunsaturated fatty acids in sheep tail fat [14] and thus to faster lipid oxidation. In fact, compounds such as pentanal, hexanal, heptanal, octanal, and nononal result from lipid oxidation. At the same time, most of the volatile compounds in cooked meat are formed by lipid reactions [39]. Pentanal, hexanal, heptanal, octanal, and 2-butanone compounds, which were found to be statistically significant, gave the highest average values in the groups containing 30% fat. However, pentanal, heptanal, and octanal compounds did not differ statistically between the groups containing 20% fat and the groups containing 30% fat. There was no statistical difference between the groups containing 10% and 20% fat in terms of hexanal, heptanal, octanal, and 2-butanone compounds (*p* > 0.05) (Table 2).

Ethyl acetate increased with the cooking time of 2 min, but increases in the cooking times to 4 and 6 min did not cause a statistical difference. Pentanal, hexanal, octanal, and 2,4-decadienal generally increased with the prolongation of the cooking time, but no statistical difference was found between the 4 and 6 min durations of the other compounds except for hexanal (*p* > 0.05). On the other hand, the level of 2-octanone, which is statistically significant among the defined ketone compounds, decreased with cooking (Table 2).

A principal component analysis (PCA) was applied to evaluate the relationships between factors (fat type, fat level, and cooking time) and volatile compounds (Figure 3). The PCA provided a good distinction between groups containing sheep tail fat (STF) and beef fat (BF) regardless of the fat content and cooking time. The groups with STF, which have different fat levels and cooking times, were on the positive side of PC1, while all BF groups were on the negative side of PC1. This result indicated that the fat type is very important for the volatile component profile of meatballs. Lipids play an important role in the development of odor and flavor of foods due to them being precursors of odor and flavor compounds or modifying the odor and flavor of other components [40].

Among the volatile compounds, aldehydes showed a higher positive correlation with groups containing STF. Furthermore, meatballs with 20% and 30% BF cooked for 4 and 6 min, and meatballs containing 10, 20, and 30% STF cooked for 6 min were placed on the negative side of PC2 and showed a positive correlation with each other (Figure 3).

### 3.4. Evaluation of the Relationship between Fat Type, Fat Level, Cooking Time, CML, and Volatile Compounds

A PCA was applied to determine the relationships between fat type, fat level, cooking time, volatile compounds, and CML (Figure 4). The first PC explained 54% of the variation. PC2 accounted for 26% of the total variance. The first two principal components explained 80% of the total variance. CML showed a positive correlation with sheep tail fat, 20 and 30% fat levels, and a cooking time of 6 min. These factors were on the positive side of PC1. In contrast, beef fat, a 10% fat level, and 0 (raw) and 2 min cooking times were on the negative side of PC1, showing a negative correlation with CML. Although CML was more affected by cooking time, it was determined that fat level and fat type also had an effect on CML formation.

In this study, the correlation between factors, volatiles, and CML was also evaluated using a heat map (Figure 5). Average linkage was used as the clustering method and Pearson correlation was used as the distance measure to illustrate the heat map. The X and Y axes represent the factors (fat type, fat level, and cooking time), and volatile compounds groups and CML, respectively. Yellow and blue were used for higher and lower correlation coefficients, respectively.

Figure 5 shows the two main clusters, and the first cluster includes ketones, furans, aldehydes, and CML, while the second cluster includes aliphatic hydrocarbons, esters, and alcohols. The first cluster was separated two subclusters. CML was generally more correlated with STF than BF. In addition, groups containing 10, 20, and 30% STF and cooked for 6 min were more correlated with CML. CML had a more positive correlation with aldehydes (Figure 5). Among the aldehydes, pentanal, octanal, 2,4 decadienal, hexanal, and heptanal were positively correlated with CML. In terms of volatile compounds, all groups containing beef fat and sheep tail fat were divided into two different clusters, regardless of the fat level and cooking time (Figure 6).

## 4. Conclusions

The use of STF at the level of 30% in meatball production significantly increased lipid oxidation. On the other hand, the BF level did not affect lipid oxidation. In contrast, the cooking time had an effect on TBARSs. Likewise, the CML content increased with increasing cooking times. However, in meatballs prepared with BF, no significant differences in the CML content were observed between cooked samples. In the presence of STF, the highest CML value was determined at the 30% fat level. In addition, fat type did not have a significant effect on CML in both raw samples and samples cooked for 2 min. When the cooking time was increased to 4 or 6 min, STF gave a higher CML content than BF. Among the volatile compounds, aldehydes were more affected by the factors examined, and these compounds exhibited a close relationship with CML. The use of sheep tail fat in meatball production, especially at the 30% level, significantly increased both CML formation and lipid oxidation.

The results showed that cooking time and the type of fat used in production are important factors for the formation of CML in meatballs. An increase in the fat level in meatballs had a positive effect on the formation of CML if the fat contained a high amount of unsaturated fatty acids (especially PUFA). These findings suggest that lipid oxidation plays an important role in the formation of CML during the cooking of beef meatballs. On the other hand, the results of this study provide new ideas for future studies on reducing CML in cooked meat products.

## Figures and Tables

**Figure 1 foods-12-02834-f001:**
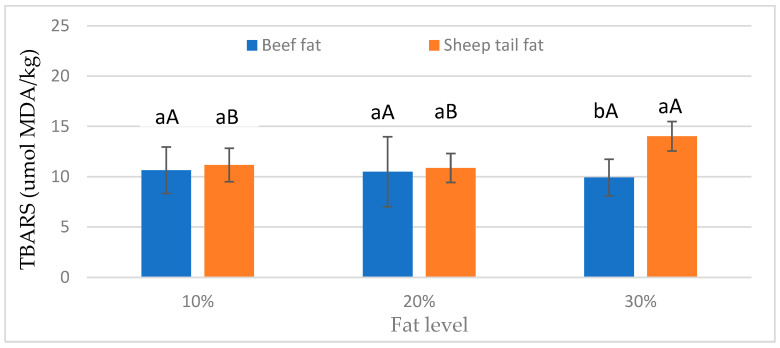
Effect of fat type × fat level interactions on TBARS values of meatballs. a, b: different small letters indicate significant differences between fat types for fat levels. A, B: different capital letters indicate significant differences between fat levels for fat types.

**Figure 2 foods-12-02834-f002:**
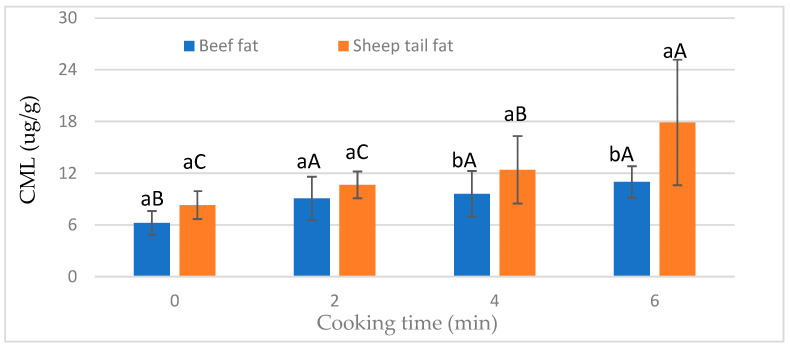
Effect of interaction of fat type and cooking time on the CML value of meatballs. a, b: different small letters indicate significant differences between fat type for cooking time. A–C: different capital letters indicate significant differences between cooking time for fat type.

**Figure 3 foods-12-02834-f003:**
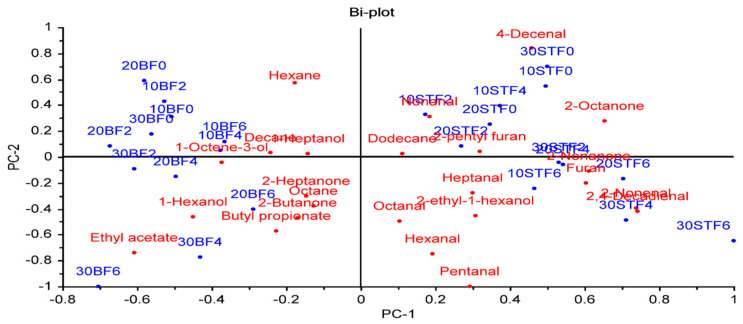
The biplot result of a principal component analysis of the relationships between factors and volatile compounds (the first number indicates the % fat level, BF: beef fat, SFT: sheep tail fat, and the last number indicates the cooking time (min)).

**Figure 4 foods-12-02834-f004:**
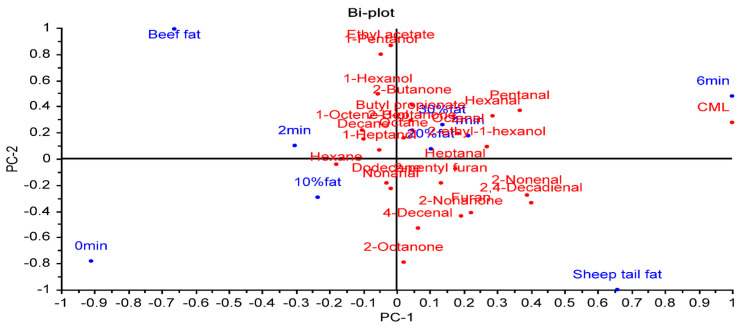
The biplot result of a principal component analysis of the relationships between factors, volatile compounds, and CML.

**Figure 5 foods-12-02834-f005:**
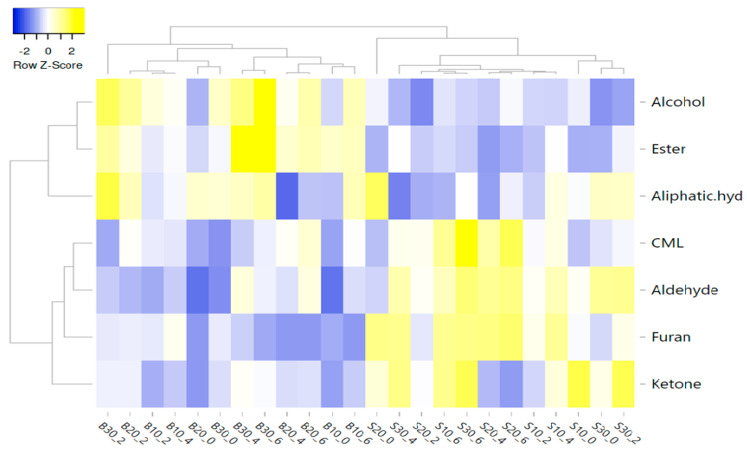
Cluster analysis of a heat map showing the relationship between factors, volatile compound groups, and CML (B: beef fat; S: sheep tail fat; 10, 20, and 30: fat level %; 0, 2, 4, and 6: cooking time (min)).

**Figure 6 foods-12-02834-f006:**
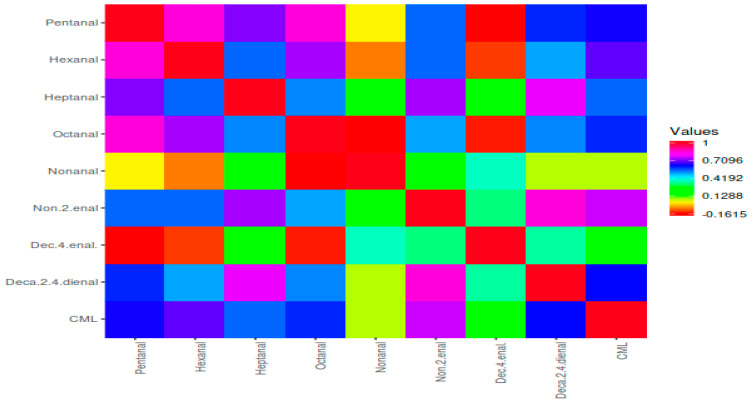
Heat map correlation matrix of CML and aldehyde compounds.

**Table 1 foods-12-02834-t001:** The effects of fat type, fat level, and cooking time on pH, TBARSs, and CML of meatballs (mean ± SD).

Factors	N	pH	TBARS (µmol MDA/kg)	CML (µg/g)
Fat type (FT)				
Beef fat	36	5.97 ± 0.12 a	10.36 ± 2.57 b	8.98 ± 2.71 b
Sheep tail fat	36	5.95 ± 0.15 a	12.02 ± 2.07 a	12.31 ± 5.44 a
Significance		ns	**	**
Fat level (FL)				
%10	24	5.94 ± 0.14 a	10.91 ± 1.98 b	10.16 ± 2.94 a
%20	24	5.95 ± 0.15 a	10.68 ± 2.61 b	11.42 ± 4.78 a
%30	24	5.98 ± 0.13 a	11.98 ± 2.64 a	10.36 ± 5.69 a
Significance		ns	*	ns
Cooking time (min)(CT)				
0 (Raw)	18	5.80 ± 0.10 c	9.88 ± 2.20 b	7.27 ± 1.81 c
2 (Rare)	18	5.93 ± 0.09 b	10.55 ± 2.17 b	9.87 ± 2.18 b
4 (Medium)	18	6.03 ± 0.09 a	11.69 ± 2.24 a	11.01 ± 3.54 b
6 (Medium–well)	18	6.06 ± 0.09 a	12.64 ± 2.46 a	14.44 ± 6.26 a
Significance		**	**	**
Interaction				
FT × FL		ns	**	ns
FT × CT		ns	ns	*
FL × CT		ns	ns	ns
FT × FL × CT		ns	ns	ns

a–c: Means marked with different letters in the same column are statistically different (*p* < 0.05), * *p* < 0.05; ** *p* < 0.01. ns: not significant.

**Table 2 foods-12-02834-t002:** The effects of fat type, fat level, and cooking time on volatile compounds of meatballs (arbitrary units × 10^6^).

Compounds			Fat Type		Fat Level (%)			Cooking Time (min)			
	KI	R	Beef fat	Sheep Tail Fat	10%	20%	30%	0	2	4	6
Hexane	600	a	3.67 ± 3.03	3.24 ± 2.46	3.44 ± 2.77	3.54 ± 3.00	3.39 ± 2.58	3.82 ± 2.67	4.66 ± 2.96	2.40 ± 2.74	2.93 ± 2.22
Octane	800	a	1.03 ± 1.51	0.67 ± 0.98	0.53 ± 0.96	0.73 ± 1.11	1.29 ± 1.61	0.74 ± 1.06	0.87 ± 1.11	0.70 ± 1.37	1.11 ± 1.58
Decane	1000	a	2.11 ± 2.91	1.18 ± 2.26	1.51 ± 2.47	1.40 ± 2.42	2.03 ± 3.02	2.01 ± 3.05	1.64 ± 2.62	1.28 ± 2.35	1.66 ± 2.62
Dodecane	1200	a	1.74 ± 2.32	2.06 ± 2.34	1.66 ± 1.80	1.21 ± 2.01	2.82 ± 2.81	2.80 ± 1.98	1.48 ± 2.43	1.15 ± 1.85	2.16 ± 2.74
Ethyl acetate	648	a	3.82 ± 1.68 a	1.26 ± 2.30 b	2.22 ± 1.65	2.18 ± 2.22	3.21 ± 3.01	1.39 ± 1.26 b	2.76 ± 2.52 a	3.21 ± 2.85 a	2.80 ± 2.34 a
Butyl propionate	952	b	1.08 ± 2.37 a	0.27 ± 0.69 b	0.58 ± 1.29	1.19 ± 0.53	1.26 ± 2.69	0.56 ± 1.31	0.26 ± 0.63	0.63 ± 2.18	1.25 ± 2.41
Pentanal	742	a	2.25 ± 1.84 b	2.92 ± 1.65 a	1.62 ± 1.36 c	2.68 ± 1.61 b	3.46 ± 1.85 a	1.00 ± 1.08 c	2.32 ± 1.53 b	3.45 ± 1.62 a	3.58 ± 1.54 a
Hexanal	849	a	2.88 ± 1.32 b	3.30 ± 1.26 a	2.20 ± 0.90 b	3.34 ± 1.22 a	3.74 ± 1.26 a	1.88 ± 0.70 c	3.05 ± 1.06 b	3.40 ± 1.17 b	4.03 ± 1.21 a
Heptanal	955	a	2.33 ± 0.91 b	3.36 ± 1.10 a	2.45 ± 0.82 b	2.86 ± 1.22 ab	3.22 ± 1.22 a	2.53 ± 0.97 b	2.53 ± 1.37 b	3.07 ± 0.97 ab	3.25 ± 1.07 a
Octanal	1044	a	3.27 ± 1.03	3.56 ± 1.02	3.20 ± 0.87 b	3.26 ± 1.06 ab	3.77 ± 1.09 a	2.56 ± 0.82 c	3.43 ± 0.82 b	3.57 ± 0.81 ab	4.08 ± 1.08 a
Nonanal	1146	a	1.88 ± 2.86	2.67 ± 2.40	2.58 ± 2.83	2.23 ± 2.85	2.01 ± 2.33	2.14 ± 2.32	2.60 ± 2.77	2.76 ± 3.02	1.59 ± 2.52
2-Nonenal	1219	b	1.13 ± 2.04 b	3.65 ± 2.29 a	2.07 ± 2.37	2.19 ± 2.36	2.91 ± 2.76	1.70 ± 2.62	1.92 ± 2.03	2.41 ± 2.30	3.52 ± 2.78
4-Decenal	1246	b	2.53 ± 3.56 b	4.47 ± 3.62 a	4.24 ± 3.83	3.00 ± 3.60	3.26 ± 3.69	3.57 ± 3.61	3.56 ± 3.77	3.73 ± 3.96	3.14 ± 3.72
2,4-Decadienal	1363	a	1.00 ± 1.79 b	3.72 ± 1.08 a	2.01 ± 1.81	2.34 ± 1.97	2.71 ± 2.23	1.69 ± 1.83 b	1.58 ± 1.79 b	2.81 ± 2.21 a	3.34 ± 1.73 a
2-Butanone	780	b	2.36 ± 2.07 a	1.43 ± 2.38 b	1.50 ± 1.90 b	1.66 ± 1.97 b	2.53 ± 2.77 a	1.26 ± 1.81	2.08 ± 2.42	1.99 ± 2.42	2.25 ± 2.41
2-Heptanone	948	b	2.28 ± 2.27	1.82 ± 2.63	2.27 ± 2.52	1.44 ± 2.16	2.44 ± 2.63	1.63 ± 2.07	2.09 ± 2.87	1.95 ± 2.15	2.52 ± 2.74
2-Octanone	1035	b	0.00 ± 0.00 b	2.50 ± 2.91 a	1.16 ± 2.60	0.94 ± 2.15	1.65 ± 3.54	2.39 ± 3.54 a	0.93 ± 1.87 b	0.94 ± 1.84 b	0.76 ± 1.68 b
2-Nonanone	1079	b	0.00 ± 0.00 b	2.34 ± 2.56 a	1.22 ± 2.29	0.69 ± 1.51	1.60 ± 2.51	0.91 ± 1.04	1.20 ± 2.07	1.36 ± 2.33	1.21 ± 2.90
1-Pentanol	835	a	2.58 ± 2.22 a	0.00 ± 0.00 b	0.85 ± 1.36	1.17 ± 1.76	1.86 ± 2.68	0.58 ± 2.16	1.35 ± 1.98	1.55 ± 1.92	1.68 ± 2.04
1-Hexanol	935	a	2.67 ± 2.10 a	1.09 ± 2.03 b	1.79 ± 1.85	1.85 ± 2.33	2.00 ± 2.47	1.13 ± 2.08	2.11 ± 2.12	2.69 ± 2.15	1.58 ± 2.32
1-Heptanol	1033	a	1.37 ± 1.63	1.00 ± 2.04	1.40 ± 2.15	1.22 ± 1.95	0.93 ± 1.38	1.05 ± 1.93	1.63 ± 1.80	0.99 ± 1.83	1.06 ± 1.90
1-Octen-3-ol	1041	b	2.16 ± 2.83 a	0.85 ± 1.60 b	1.37 ± 2.18	1.42 ± 2.07	1.71 ± 2.89	2.04 ± 2.92	1.16 ± 1.89	0.97 ± 1.82	1.85 ± 2.71
2-Ethyl-1-hexanol	1084	b	1.95 ± 2.70	2.65 ± 2.79	2.36 ± 2.81	1.94 ± 2.88	2.60 ± 2.63	1.70 ± 2.35	1.94 ± 2.53	1.63 ± 2.16	3.92 ± 3.36
2-Pentyl furan	1022	b	0.85 ± 2.36	2.12 ± 3.14	2.02 ± 3.36	0.82 ± 2.05	1.61 ± 2.91	0.88 ± 2.06	1.78 ± 2.93	1.70 ± 3.33	1.59 ± 2.99
Furan	1107	b	0.50 ± 1.20 b	2.74 ± 3.18 a	1.11 ± 1.36	2.23 ± 3.51	1.52 ± 2.58	1.50 ± 2.64	1.08 ± 1.29	2.23 ± 3.57	1.68 ± 2.64

Results are expressed in arbitrary area units (×10^6^); KI: Kovats index calculated for a DB-624 capillary column (30 m × 0.25 mm × 1.4 μm film) installed on a gas chromatograph equipped with a mass selective detector. R: reliability of identification, a: mass spectrum and retention time identical to an authentic sample; b: mass spectrum and Kovats index from the literature. a–c: means marked with different letters in the same row in same section are statistically different (*p* < 0.05).

## Data Availability

The data presented in this study are available on request from the corresponding author.

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
