# Peer review of "The Effects of Sheep Tail Fat, Fat Level, and Cooking Time on the Formation of Nε-(carboxymethyl)lysine and Volatile Compounds in Beef Meatballs"

_foods, 2023, doi:10.3390/foods12152834_

Round 1

Reviewer 1 Report

Introduction

Write a paragraph comparing the beef fat and sheep tail fat. What are the similarities and differences in chemical composition (especially fatty acids)

Section: 2.2. Meatball production and cooking

Please show the recipe of meatball in Table?

Why cooking time is maximum 6 min? Why not 8-10 min?  Which criteria is used for choosing 6 min? 

In Results and discussion section

Please write more explanations on below comments:

What is a main finding of this research? 

How much fat need to add into meatball formulation for obtaining good quality meatball? 30%? 

Author Response

Reviewer 1

Introduction

Write a paragraph comparing the beef fat and sheep tail fat. What are the similarities and differences in chemical composition (especially fatty acids)

-Added

Section: 2.2. Meatball production and cooking

Please show the recipe of meatball in Table?

- Only salt was used in the production. No spices were added during the production of meatballs. While the salt ratio (1.5%) was constant, the fat ratio was used as stated in the method section. Therefore, it is not given in a table.

 Why cooking time is maximum 6 min? Why not 8-10 min?  Which criteria is used for choosing 6 min? 

- The durations were determined based on four different cooking intensity (raw, rare, medium, medium well) with preliminary trials. The cooking intensity could have been further increased. However, it is not within the scope of this project.

In Results and discussion section

Please write more explanations on below comments:

What is a main finding of this research? 

-Added

How much fat need to add into meatball formulation for obtaining good quality meatball? 30%? 

- Fat is a major ingredient in terms of taste, texture and flavor. In the study, it was determined that the fat ratio had no significant effect on CML. In contrast,  the interaction of fat type x cooking time were more effective.

Revised

Reviewer 2 Report

In this manuscript, the authors summarized and discussed the effects of sheep tail fat, fat level and cooking time on the formation of Nε-(carboxymethyl)lysine and volatile compounds in beef meatballs. This work provides new insight and opinion into the development of AGE formation in meatballs. The manuscript is well - organized and clearly stated. However, there still have some issues need to check. I would suggest accepting it after the following concerns are addressed.

1.     Introduction. The background about Maillard reaction information should be more illustration.

2.     The AGEs have adverse effects on human health should be separated paragraph, please refer this reference (Critical Reviews in Food Science and Nutrition. 2023, Doi: 10.1080/10408398.2023.2213768).

3.     The data analyzation in Table 1 should be confirmed.

4.     “2.5. Nε-(carboxymethyl)lysine (CML) analysis”. The AGEs detection method should be introduced and analyzed .

5. There is a problem with the table header in Table 2.

6. The size of the graph should be as uniform as possible.

7. The references should be updated. Some closedly related and new references should be added and reviewed.

8. The main contribution is not clear. Please highlight it.

9. In the conclusion, the relationship between fat type, fat level, cooking time, CML and volatile compounds was appropriately added.

10. The reference should be updated.

In this manuscript, the authors summarized and discussed the effects of sheep tail fat, fat level and cooking time on the formation of Nε-(carboxymethyl)lysine and volatile compounds in beef meatballs. This work provides new insight and opinion into the development of AGE formation in meatballs. The manuscript is well - organized and clearly stated. However, there still have some issues need to check. I would suggest accepting it after the following concerns are addressed.

1.     Introduction. The background about Maillard reaction information should be more illustration.

2.     The AGEs have adverse effects on human health should be separated paragraph, please refer this reference (Critical Reviews in Food Science and Nutrition. 2023, Doi: 10.1080/10408398.2023.2213768).

3.     The data analyzation in Table 1 should be confirmed.

4.     “2.5. Nε-(carboxymethyl)lysine (CML) analysis”. The AGEs detection method should be introduced and analyzed.

5. There is a problem with the table header in Table 2.

6. The size of the graph should be as uniform as possible.

7. The references should be updated. Some closedly related and new references should be added and reviewed.

8. The main contribution is not clear. Please highlight it.

9. In the conclusion, the relationship between fat type, fat level, cooking time, CML and volatile compounds was appropriately added.

10. The reference should be updated.

Author Response

Reviewer 2

In this manuscript, the authors summarized and discussed the effects of sheep tail fat, fat level and cooking time on the formation of Nε-(carboxymethyl)lysine and volatile compounds in beef meatballs. This work provides new insight and opinion into the development of AGE formation in meatballs. The manuscript is well - organized and clearly stated.

-Thank you

However, there still have some issues need to check. I would suggest accepting it after the following concerns are addressed.

  1. Introduction. The background about Maillard reaction information should be more illustration.

-Added

  1. The AGEs have adverse effects on human health should be separated paragraph, please refer this reference (Critical Reviews in Food Science and Nutrition. 2023, Doi: 10.1080/10408398.2023.2213768).

-Added

  1. The data analyzation in Table 1 should be confirmed.

Thank you. Checked.

  1. “2.5. Nε-(carboxymethyl)lysine (CML) analysis”. The AGEs detection method should be introduced and analyzed .

Since CML is an indicator for AGEs, only CML is used in the title.

  1. There is a problem with the table header in Table 2.

-Revised

  1. The size of the graph should be as uniform as possible.

-Revised

  1. The references should be updated. Some closedly related and new references should be added and reviewed.

-Revised

  1. The main contribution is not clear. Please highlight it.

-Revised

  1. In the conclusion, the relationship between fat type, fat level, cooking time, CML and volatile compounds was appropriately added.

-Thank you 

  1. The reference should be updated.

-Revised

Reviewer 3 Report

As it was mentioned by the Authors there still not many articles dedicated to meat and Maillard reaction product formation, especially on the advanced stage. However, the concept is proper, in my opinion, some corrections are needed.

Line 51: There should be an explanation of how these parameters could be connected. Why the Authors used this concept?

Line 28: I suggest writing: "early, advanced and last stages". AGEs are a part of the advanced stage. 

Line 30: instead of "this reaction", convert it to "these reactions" Maillard process is not one reaction as was mentioned above by the Authors.

Line 31: please check the grammar of this sentence.

Line 41: maybe it is better to write "established" instead of "accepted"

Line 44: try to be consequent and write AGEs all the time.

Line 43: the Authors wrote "cooking degree" or do you mean "cooking temperature" or "cooking temperature"?

Line 71: how was the temperature controlled in the whole experiment?

Lack of subchapter about chemicals and materials used for analysis. Moreover, in the methods, there are no references to literature. In the volatiles analysis, the Authors mentioned that "standard substances were 123 used for the identification of compounds". The list of standards and producers has to be provided.

Line 137: "heatmapper" is a program, if yes the info about it should be added.

Line 172: please check the subtitle for Figure 1. Moreover, Table 1 should be added just below the part of the text describing it.

Line 178: " This type of fat also showed the highest mean TBARS value (Table 1)"?

Table 1 what means "N"? In the lexicon "raw"= "rare" could you find better wording to this?

I advise checking statistics for volatiles in Table 1  in the "cooking time", for example, hexane, octane etc.

Table 3. Provide full name for the abbreviations, cause it was not used before.

Line 290: better to write "groups of volatile compounds", "volatiles"

in the discussion part, there is a lack of possible mechanisms which can provide volatiles or CML formation, and which one could be more favourable.

As it was mentioned by the Authors there still not many articles dedicated to meat and Maillard reaction product formation, especially on the advanced stage. However, the concept is proper, in my opinion, some corrections are needed.

Line 51: There should be an explanation of how these parameters could be connected. Why the Authors used this concept?

Line 28: I suggest writing: "early, advanced and last stages". AGEs are a part of the advanced stage. 

Line 30: instead of "this reaction", convert it to "these reactions" Maillard process is not one reaction as was mentioned above by the Authors.

Line 31: please check the grammar of this sentence.

Line 41: maybe it is better to write "established" instead of "accepted"

Line 44: try to be consequent and write AGEs all the time.

Line 43: the Authors wrote "cooking degree" or do you mean "cooking temperature" or "cooking temperature"?

Line 71: how was the temperature controlled in the whole experiment?

Lack of subchapter about chemicals and materials used for analysis. Moreover, in the methods, there are no references to literature. In the volatiles analysis, the Authors mentioned that "standard substances were 123 used for the identification of compounds". The list of standards and producers has to be provided.

Line 137: "heatmapper" is a program, if yes the info about it should be added.

Line 172: please check the subtitle for Figure 1. Moreover, Table 1 should be added just below the part of the text describing it.

Line 178: " This type of fat also showed the highest mean TBARS value (Table 1)"?

Table 1 what means "N"? In the lexicon "raw"= "rare" could you find better wording to this?

I advise checking statistics for volatiles in Table 1  in the "cooking time", for example, hexane, octane etc.

Table 3. Provide full name for the abbreviations, cause it was not used before.

Line 290: better to write "groups of volatile compounds", "volatiles"

in the discussion part, there is a lack of possible mechanisms which can provide volatiles or CML formation, and which one could be more favourable.

Author Response

Reviewer 3

As it was mentioned by the Authors there still not many articles dedicated to meat and Maillard reaction product formation, especially on the advanced stage. However, the concept is proper, in my opinion, some corrections are needed.

Line 51: There should be an explanation of how these parameters could be connected. Why the Authors used this concept?

-Revised

Line 28: I suggest writing: "early, advanced and last stages". AGEs are a part of the advanced stage. 

-Revised

Line 30: instead of "this reaction", convert it to "these reactions" Maillard process is not one reaction as was mentioned above by the Authors.

-Revised

Line 31: please check the grammar of this sentence.

-Revised

Line 41: maybe it is better to write "established" instead of "accepted"

-Revised

Line 44: try to be consequent and write AGEs all the time.

-Revised

Line 43: the Authors wrote "cooking degree" or do you mean "cooking temperature" or "cooking temperature"?

-Revised

Line 71: how was the temperature controlled in the whole experiment?

-The cooking process was carried out on a hot plate preheated at 180°C before cooking and its temperature was controlled by measurement with a digital thermocouple (Testo 926, Testo, Titisee-Neustadt, Germany).

Lack of subchapter about chemicals and materials used for analysis. Moreover, in the methods, there are no references to literature. In the volatiles analysis, the Authors mentioned that "standard substances were 123 used for the identification of compounds". The list of standards and producers has to be provided.

-Added

Line 137: "heatmapper" is a program, if yes the info about it should be added.

-Added

Line 172: please check the subtitle for Figure 1. Moreover, Table 1 should be added just below the part of the text describing it.

-Revised

Line 178: " This type of fat also showed the highest mean TBARS value (Table 1)"?

-Revised

Table 1 what means "N"? In the lexicon "raw"= "rare" could you find better wording to this?

-N = number of samples.

 There are 3 factors in the research. The number of repetitions is 3. In other words, the production was made  for 3 times.

 N value for fat type: 3x4x3=36

N value for fat level=2x4x3=24

N value for cooking time=2x3x3=18

Since the term "rare" is used in the literature, no better expression for this word could be found.

I advise checking statistics for volatiles in Table 1  in the "cooking time", for example, hexane, octane etc.

The values have been recontrolled. This study has a factorial experiment plan. Therefore, standard deviations can show a wide variation.

Table 3. Provide full name for the abbreviations, cause it was not used before.

Table 3 is not available in this study. There are explanations for Figure 3.

Line 290: better to write "groups of volatile compounds", "volatiles"

-Revised

in the discussion part, there is a lack of possible mechanisms which can provide volatiles or CML formation, and which one could be more favourable.

-Revised

Reviewer 4 Report

This study was performed to determine the effects of fat type, level, and cooking time on the Maillard reaction products and volatile compounds in beef meatballs. The objective is reasonable, but some information on the experimental settings should be improved.

 L43-52 In introduction, detailed information on why sheep tail fat was considered should be addressed.

L55 Please describe the scientific name of beef muscles used and the location of collected intermuscular fat, since the chemical composition considerably varies depending upon locational difference.

L65 Delete “t” in parentheses.

L68-70 The core temperature of samples at each cooking temperature should be described.

L70-71 Considering the shape of meatballs, the cooking procedure should be described in more detail.

L101 Carboxymethyl -> carboxymethyl

L106 Use upper letter regarding R2.

L141 It would be better to change the order of Fig. 1 and Table 1.

L147 Please check the space in “cooking can”.

L155 It would be better to revise “effective” for renderability.

L162 Please consistently use one of Figure and Fig. in entire manuscript.

The typo errors and readability should be doubled-checked.

Author Response

Reviewer 4

This study was performed to determine the effects of fat type, level, and cooking time on the Maillard reaction products and volatile compounds in beef meatballs. The objective is reasonable, but some information on the experimental settings should be improved.

 L43-52 In introduction, detailed information on why sheep tail fat was considered should be addressed.

Revised

L55 Please describe the scientific name of beef muscles used and the location of collected intermuscular fat, since the chemical composition considerably varies depending upon locational difference.

 Fats between the muscles of carcass were used. No specific muscle was chosen here. For this reason, it is indicated as intermuscular fat.

L65 Delete “t” in parentheses.

Removed.

L68-70 The core temperature of samples at each cooking temperature should be described.

The durations were determined based on four different cooking intensity (raw, rare, medium, medium well) with preliminary trials. The cooking intensity could have been further increased. However, it is not within the scope of this project. 

L70-71 Considering the shape of meatballs, the cooking procedure should be described in more detail.

The thickness of the meatballs is 1.5 cm thick and the diameter is 7.5 cm diameter.  Therefore, their shape was suitable for cooking on a flat surface. In this study, a hot plate was used for cooking and each surface was turned depending on the cooking time.

L101 Carboxymethyl -> carboxymethyl

 Changed

L106 Use upper letter regarding R2.

Revised

L141 It would be better to change the order of Fig. 1 and Table 1.

 Ok. Changed

L147 Please check the space in “cooking can”.

 Thank you. Corrected

L155 It would be better to revise “effective” for renderability.

I am sorry. I could not change it because I could not understand the mean of "renderability"

L162 Please consistently use one of Figure and Fig. in entire manuscript.

Thank you. Corrected

The typo errors and readability should be doubled-checked.

Thank you. Checked.

Round 2

Reviewer 1 Report

All notes and comments are corrected and added

The manuscript is ready for publication

Reviewer 2 Report

The author has responded to the reviewer's comment point by point. It can be accepted in the current revision.

The author has responded to the reviewer's comment point by point. It can be accepted in the current revision.